# Chemical, Nutritional and Biological Evaluation of a Sustainable and Scalable Complex of Phytochemicals from Bergamot By-Products

**DOI:** 10.3390/molecules28072964

**Published:** 2023-03-26

**Authors:** Larissa Della Vedova, Francesca Gado, Taynara A. Vieira, Núbia A. Grandini, Thiago L. N. Palácio, Juliana S. Siqueira, Marina Carini, Ezio Bombardelli, Camila R. Correa, Giancarlo Aldini, Giovanna Baron

**Affiliations:** 1Department of Pharmaceutical Sciences, University of Milan, Via Mangiagalli 25, 20133 Milan, Italy; 2Medical School, Sao Paulo State University (Unesp), Botucatu 18618-687, Brazil; 3Plantexresearch Srl, 20122 Milan, Italy

**Keywords:** bergamot, polyphenols, antioxidant, anti-inflammatory, NfkB, metabolic syndrome

## Abstract

The present paper reports a sustainable raw material obtained from the by-products derived from the industrial production of bergamot (*Citrus* × *Bergamia* Risso & Poiteau) essential oils. The procedure to obtain the raw material is designed to maintain as much of the bioactive components as possible and to avoid expensive chemical purification. It consists of spray-drying the fruit juice obtained by squeezing the fruits, which is mixed with the aqueous extract of the pulp, i.e., the solid residue remained after fruit pressing. The resulting powder bergamot juice (PBJ) contains multiple bioactive components, in particular, among others, soluble fibers, polyphenols and amino-acid betaines, such as stachydrine and betonicine. LC-MS analysis identified 86 compounds, with hesperetin, naringenin, apigenin and eridictyol glucosides being the main components. In the second part of the paper, dose-dependent anti-inflammatory activity of PBJ and of stachydrine was found, but neither of the compounds were effective in activating Nrf2. PBJ was then found to be effective in an in vivo model of a metabolic syndrome induced by a high-sugar, high-fat (HSF) diet and evidenced by a significant increase of the values related to a set of parameters: blood glucose, triglycerides, insulin resistance, systolic blood pressure, visceral adipose tissue and adiposity index. PBJ, when given to control rats, did not significantly change these values; in contrast, they were found to be greatly affected in rats receiving an HSF diet. The in vivo effect of PBJ can be ascribed not only to bergamot polyphenols with well-known anti-inflammatory, antioxidant and lipid-regulating effects, but also to the dietary fibers and to the non-phenolic constituents, such as stachydrine. Moreover, since PBJ was found to affect energy homeostasis and to regulate food intake, a mechanism on the regulation of energy homeostasis through leptin networking should also be considered and deserves further investigation.

## 1. Introduction

Bergamot (botanical name under the Vienna Code of 2006, *Citrus* × *bergamia* Risso & Poiteau [1]) is a citrus fruit that grows almost exclusively in southern Italy in a restricted area of the Calabrian coast, and is characterized by a unique profile in flavonoids, including flavanones (such as naringenin, hesperetin and eriodictyol glycosides), flavones, (apigenin, luteolin, chrysoeriol and diosmetin glycosides) and their 3-hydroxy-3-methyl-glutaryl (HMG) derivatives [2,3]. Bergamot fruits were initially grown for use in the perfume, cosmetic, food and confectionery industries [4], but it is now known that bergamot phytochemicals and, in particular, flavonoids possess several pharmacologically beneficial effects, including hypolipemic, hypoglycemic, antioxidant and anti-inflammatory activities, resulting in beneficial actions as found in pre-clinical and intervention studies [5,6,7]. Some molecular mechanisms of bergamot flavonoids and the molecules that are responsible for biological effects have been partially clarified, as reviewed by Sadeghi-dehsahraei et al. [7]. The molecular mechanisms involve, among others, enzyme inhibition, modulation of phosphorylation and gene regulation. As an example, the lipid-lowering effect can be ascribed to buteridine and melitidine, which contain a 3-hydroxy-3-methylglutaryl moiety as a statin, acting as competitive inhibitors of HMG-CoA reductase, a key enzyme for cholesterol synthesis [8,9]. In addition to reducing cholesterol, bergamot polyphenols also decrease triglycerides (TG) plasma levels and TG accumulation in the liver by reducing the activity of the enzymes involved in TG synthesis, such as phosphatidate phosphohydrolase [10]. Regarding the well-established anti-inflammatory activity of bergamot flavonoids, Risitano et al. [11] have found that they inhibit both gene expression and secretion of LPS-induced pro-inflammatory cytokines (IL-6, IL-1b, TNF-a) by a mechanism involving the inhibition of NF-kB activation. Recently, Mugeri et al. have reported the potential anti-leukemic effect of a flavonoid-rich extract of bergamot juice in THP-1 cells due to a reduced AKT phosphorylation, which plays a pivotal role in bergamot polyphenols-induced cell cycle arrest and apoptosis in THP-1 [12]. Hence, bergamot contains several phytochemicals acting on different biological targets.

It is quite clear that the beneficial effects of bergamot are given by the combination of the different biological activities evoked by multiple components, which act through different molecular mechanisms.

In general, it is now well accepted that the overall activities of the phytochemicals promote a synergistic action, which not only occurs among compounds of the same chemical classes, such as bergamot flavonoids, but also among compounds that are not structurally related to each other. As an example, it has been clearly shown that the anti-inflammatory action of polyphenols is enhanced when they are combined with compounds belonging to other chemical classes and deriving from the same or different fruit [13]. In line with this principle, a bergamot polyphenol extract complex containing both flavonoids and pectins has been reported to efficiently induce a combination of weight loss and insulin sensitivity effects, together with a robust reduction of atherosclerosis risk. The effects of the mixture on weight loss are characteristic of the mixture itself, since bergamot flavonoids counteract dyslipidemia and hyperglycemia but fail to induce a significant weight loss [14].

Based on the view that natural ingredients have a synergistic action when present in a mixture and also taking into account that limiting the purification steps is a clear industrial advantage in terms of sustainability and cost, we here propose the chemical, nutritional and biological evaluation of a non-purified raw material obtained from bergamot by-products; in particular, the dry extract is obtained by spray-drying the fruit juice, which is obtained by squeezing the fruits and is mixed with the aqueous extract of the pulp, i.e., the solid residue remained after fruit pressing. Bergamot fruit is used especially for the extraction of its essential oil from the peel (by cold pressing), while the bergamot juice and the remaining pulp are considered a by-product of the essential oil production [15,16]. In more detail, the essential oil derived from bergamot is highly valued in the pharmaceutical industry for its antibacterial and antiseptic properties, as well as in the cosmetic industry for producing perfumes, body lotions and soaps. Moreover, it is commonly used in the food industry as an aroma in sweets, liquors and tea. Conversely, the bitter taste of bergamot juice has limited its use in food production and is considered a by-product of essential oil production. Another relevant by-product of the bergamot chain is represented by the pulp, which can find some application as an energy supplement in ruminant diets [16], especially in bergamot-producing regions. As a result, the disposal of bergamot juice and pulp poses a significant economic and environmental challenge for essential oil processing industries [17,18].

We here report the preparation of an extract named PBJ (powder bergamot juice), which is obtained by a scalable industrial process of bergamot by-produtcs (fruit juice and pulp) that avoids chemical purification steps, thus maintaining the bioactive ingredients of bergamot (soluble fibers, polyphenols and small non-phenolic hydrophilic derivatives) in their natural integrity and combination. The first part of the paper describes the procedure for a scalable preparation of PBJ, followed by a full chemical and nutritional characterization. In the second part of the paper, the biological activity of PBJ is evaluated in cell models to test its anti-inflammatory and antioxidant activity, and in an in vivo rat model of metabolic syndrome induced by a high caloric intake.

## 2. Results

### 2.1. PBJ Composition

Table 1 reports the relative percentage nutritional composition of PBJ, which contains more than 55% of carbohydrates with a relative content of glucose and fructose of 12.3% and 10.6%, respectively. The total dietary fiber accounted for 19.4%, of which the soluble fraction was more than 18%. The relative content of the protein was less than 6% and the lipid fraction is present in a negligible amount, 0.24% of which more than 50% is represented by PUFAs.

Identification of polyphenols and their semi-quantitative analyses were performed by LC-ESI-MS, while quantitative analyses of the most abundant classes by HPLC-UV-DAD were made using an external calibration method.

Polyphenols, as determined by HPLC-UV analysis, account for 10–13%, and their qualitative composition performed by HPLC coupled with a high-resolution MS analyser is summarized in Table 2. A total of 86 compounds were identified: 83 in negative ion mode, 56 in positive ion mode and 51 confirmed in both polarities. Figure 1 shows the total ion current (TIC) of PBJ acquired in the negative ion mode. The numbers on the chromatogram peaks refer to the identified compounds listed in Table 2. The relative abundance of the identified metabolites was calculated by measuring the relative AUC of the peaks reconstituted by setting the molecular ion of the metabolite as filter ion and a tolerance of 5 ppm. The semi-quantitative method clearly suffers from the limitation due to a different ionization efficiency of the detected metabolites, but it gives a draft indication of the main components. The results indicate that among the polyphenolic components, hesperitin glucosides are the most abundant (~40%), followed by naringenin (~30%), eriodictyol (~21%) and apigenin (~4%) glucosides. Besides polyphenols, other hydrophilic components contained in PBJ include citric acid, quinic acid and stachydrine.

Hence, based on the analytical data, PBJ represents a valuable source of bioactive ingredients with well-established health activities and includes soluble fibers, polyphenols and stachydrine. Bearing in mind the bioactive components of PBJ, we then moved to evaluate the efficacy of PBJ in in vitro models first, and then in pre-clinical studies.

### 2.2. In Vitro Studies

#### 2.2.1. Anti-Inflammatory Activity

We first evaluated the anti-inflammatory effects of PBJ in a cell model of inflammation induced by TNF-α by measuring the NfkB nuclear translocation. An MTT assay was first evaluated to test the cell toxicity of PBJ, which did not affect cell viability up to the highest tested dose of 250 µg/mL. In the first set of experiments, PBJ was not effective in reducing NfkB nuclear translocation up to the highest tested dose of 250 µg/mL, which corresponds to a content of polyphenols of 32 µg/mL, as determined by considering a relative content of polyphenols in PBJ of almost 13% (Figure 2A). The data do not provide an apparent explanation, since in previous experiments [3], we found that isolated bergamot polyphenols showed a significant anti-inflammatory activity at a dose as low as 10 µg/mL. Moreover, other anti-inflammatory constituents of PBJ, such as amino-acid betaines (stachydrine and betonicine), should potentiate the anti-inflammatory activity.

These apparently contradictory results can be explained by considering that, in PBJ, the polyphenols and other small molecules are embedded in the polymeric structure of the fibers or complexed with macromolecules and are thus not available in the cell medium for the activity. This is clearly not the case when testing polyphenols isolated from the natural media through solvent extraction and then purified by chromatography, since in this case, they are fully bioavailable for the activity. Such in vitro conditions are clearly far from those occurring in vivo, where the polyphenols are released from the polymeric matrix upon soluble fiber digestion from gut microbiota [19]. With the aim of testing this explanation, the PBJ was extracted in a hydro–alcoholic mixture and under sonication to release the small molecules from the polymeric matrix. The percentage extraction yield was 77.9 ± 1.2% and the qualitative composition of the polyphenols was maintained, as determined by LC-ESI-MS.

The fraction obtained by sonication/extraction was found to be effective in a concentration range of 10–250 µg/mL, which demonstrates very good activity, taking into consideration the relative content of the bioactive components (Figure 2B). In particular, by considering the yield of extraction after hydroalcoholic and sonication treatment and the relative amount of polyphenols in PBJ, the range of activity expressed as polyphenol content starts from 1.62 μg/mL to increase dose-dependently at 16.2 and 40 μg/mL. Stachydrine was also found to dose-dependently reduce the TNF-alfa mediated inflammatory response. The efficacy was already significant at 1 nmoles/mL corresponding to 0.143 µg/mL, an amount which is contained in 20 μg/mL of bergamot, and which should contribute to the activity (Figure 2C).

#### 2.2.2. Radical Scavenging and Cellular Antioxidant Activity

The results of the direct radical scavenging of PBJ towards the stable radical DPPH is depicted in Table 3. The IC_50_ is 233 µg/mL, which corresponds to 30 µg/mL when expressed as polyphenols, a potency which overlaps that observed for polyphenols isolated from bergamot leaves and fruit.

We then tested the efficacy of PBJ to potentiate the cell antioxidant enzymes through the nuclear factor erythroid 2–related factor 2 (Nrf2) pathway, such as we have already found for polyphenols from different sources [20,21,22]. Figure 3 reports the dose-dependent effect of PBJ on Nrf2 activation after 18 h of incubation in a concentration range between 10 and 250 µg/mL. The potency of PBJ as an Nrf2 activator was very weak, being effective only at the highest dose and after incubation for 18 hrs. We then tested the efficacy of PBJ after alcohol extraction and sonication, but the potency did not change significantly. We then tested the Nrf2 activation of stachydrine, which was also found not to be effective in the dose-range found active in the anti-inflammatory model. Taken together, the data well indicate that the anti-inflammatory activity of PBJ, as well as of stachydrine, is not related to the Nrf2 pathway.

### 2.3. In Vivo Studies

#### 2.3.1. Nutritional Intake

Figure 4 shows the parameters related to the nutritional intake. It is possible to note that the HSF and HSF+PBJ groups were fed less compared to the control and control plus PBJ, respectively (Figure 4A). At the same time, it is possible to observe that the HSF group had higher water intake when compared to control and HSF+PBJ (Figure 4B,C), and an increase in caloric intake compared to the control group (Figure 4B).

#### 2.3.2. Metabolic Syndrome Parameters

The metabolic syndrome parameters are presented in Figure 5. It can be seen that the HSF diet used in this study promoted obesity and significant metabolic changes when compared to the control group. This fact is evidenced by a significant increase of the values related to a set of parameters involved in the metabolic syndrome: blood glucose, triglycerides, insulin resistance (TyG), systolic blood pressure, visceral adipose tissue and adiposity index. PBJ, when given to the control rats, did not significantly change these values, which, in contrast, were found to be greatly affected in the rats receiving an HSF diet. The HSF+PBJ group had significantly improved triglycerides, TyG, systolic blood pressure, visceral adipose tissue and adiposity index in comparison to the HSF group, as shown in Figure 5.

## 3. Discussion

A PBJ industrial preparation was designed to maintain the main functional ingredients found in bergamot juice (soluble fibers, polyphenols and non-polyphenolic small molecules) so that they can act in cooperation and with a pleiotropic mechanism. Moreover, the industrial procedure is sustainable and has a reduced economic impact since it avoids several purification steps.

PBJ was first found effective in reducing the cell-inflammatory response mediated by NfkB nuclear translocation induced by TNF-alfa. To show the activity, PBJ must be extracted and sonicated and this demonstrates that in the crude matrix, polyphenols and, in general, anti-inflammatory small molecules are embedded in the matrix and are thus not available for the biological action. To evidence the activity in in vitro conditions, polyphenols need to be released, which occurs by solvent extraction and sonication. In the in vivo experiments, the enzymatic degradation of the matrix and, in particular, of the soluble fibers, which occurs in the gastro-intestinal tract, makes the small molecules available for the biological activity.

There is clearly an advantage with polyphenols/small components when embedded in the natural matrix, which is that they are stable and less susceptible to air and, in general, to oxidative degradation. Moreover, natural matrices can affect the bioavailability of phytochemicals and stimulate their microbiota-driven conversion to bioactive metabolites, as already demonstrated for procyanidins, which are metabolized to bioactive valerolactone derivatives [23].

The molecular mechanisms causing the in vitro and in vivo activity of PBJ can be explained by considering the mixture of bioactive components that are characteristic of PBJ, including polyphenols, small non-phenolic constituents, such as amino-acid betaines (stachydrine and betonicine) and soluble fibers. Polyphenolic constituents can represent an active fraction concurring with the anti-inflammatory effect through a pleiotropic mechanism. The dose with a significant anti-inflammatory efficacy was found not effective in activating the Nrf2 pathway, thus suggesting the involvement of other mechanisms beyond those promoted by the antioxidant effect as a consequence of Nrf2 activation. Many abundant compounds identified in PBJ have a well-established antioxidant and anti-inflammatory activity through a multi-target mechanism. A main characteristic fraction of PBJ is represented by seven different hesperetin glucosides accounting for 40% of the relative ionic abundance, as calculated by measuring the sum of the AUCs of the corresponding ion peaks with respect to the total AUCs of all the detected peaks. Hesperetin is a 4′-methoxy derivative of eriodictyol, a flavanone, whose glucoside derivatives are found in citrus fruits. Hesperidin derivatives have potent anti-inflammatory activity, as demonstrated in several animal models, as well as in humans [24]. The absence of an ortho- or para-diphenol moiety suggests that these compounds are ineffective in Nrf2 activation, while other molecular mechanisms explaining the anti-inflammatory activity have been proposed, including a strong inhibition of MAP kinases and expression of p-ERK and p38, thus reducing the levels of inflammatory cytokines [25,26].

Naringenin and glycosides (30% of ionic abundance) are other important components present in PBJ. Naringenin, like hesperidin, is a flavanone devoid of a moiety effective as an Nrf2 activator, but with well-established anti-inflammatory activity with a pleiotropic mechanism, including inhibition of protein kinases; free radical production; inflammatory enzymes, including PLLA2, COS and LOX; and modulation transcription factors, including NfkB, gata-3 and STAT-6 [27].

Similar consideration can be drawn for apigenin derivatives (4%), which act by decreasing the levels of pro-inflammatory cytokines and inflammatory mediators by downregulating the MAPK, NfkB and Jak/STAT signaling pathways [28].

The action of PBJ polyphenols on Nrf2 should also be considered, although it is not involved as a primary mechanism, as evidenced by our in vitro tests. Nrf2 activation can be supported by some polyphenol derivatives found in PBJ with an ortho-diphenol moiety, including eriodictyol glycosides.

The anti-inflammatory activity of bergamot could also be ascribed to other non-phenolic constituents, such as stachydrine, which comprises a significant relative amount (more than 0.7%) and acts through a mechanism not involving the Nrf2 pathway. To date, stachydrine has demonstrated various bioactivities for the treatment of fibrosis, cardiovascular diseases, cancers, uterine diseases, brain injuries and inflammation [29]. Many studies have demonstrated that stachydrine has strong antifibrotic properties (on various types of fibrosis) inhibiting extra cellular matrix deposition and decreasing inflammatory and oxidative stress through multiple molecular mechanisms (including TGF-β, ERS-mediated apoptosis, MMPs/TIMPs, NF-κB and JAK/STAT). Jung et al. have recently found that the anti-inflammatory effect of stachydrine may be associated with IL-10 signaling, including the AMPK/HO-1-dependent pathway [30].

The effect of bergamot polyphenols in reducing metabolic syndrome parameters is a well-established fact, as demonstrated in pre-clinical and clinical studies and recently reviewed by Carresi et al. [5]. We here confirm these data by using PBJ, a sustainable product derived from the bergamot fruit, rich in bioactive components and containing polyphenols and other small bioactive molecules, which are embedded in the natural matrix. The in vivo effect of PBJ on metabolic syndrome parameters can be ascribed to its radical-scavenging and anti-inflammatory activities, as found in in vitro conditions, but other mechanisms should also be considered. The hypolipemic effects of PBJ can be explained by the modulation of the activity of some enzymes responsible for cholesterol esterification reactions and lipid trafficking.

Very recently, in addition to polyphenols, other bergamot constituents, such as the soluble (SDF) and insoluble fiber fractions, have attracted scientific interest [31]. SDF is characterized by a porous structure, which enhances the water and oil holding capacity, as well as cholesterol and glucose absorption. Soluble fibers have been effective in reducing metabolic syndrome parameters. In diabetic db/db mice, bergamot fibers were found to be effective in regulating glycemia, enhancing insulin sensitivity, regulating glucose homeostasis and effectively controlling blood lipid levels [31].

Finally, it should be considered that the in vivo effects of bergamot can also be explained by the fact that the HSF+PBJ group received less water and, hence, a lower sugar intake in respect to HSF. The high sugar content in the water is responsible for the high caloric intake in HSF, promoting weight gain, which was significantly reduced in the PBJ-treated rats. The reduced water/sugar intake can be explained by the positive effect of PBJ on energy homeostasis, food intake regulation and energy expenditure. Several mechanisms can be considered to explain this effect, opening a novel future research on this aspect. Among these, there is a possible regulating effect on leptin signalling, leptin being a hormone secreted mainly from the white adipose tissue, which is the main messenger that carries information about peripheral energy stores to the hypothalamus. Leptin is unable to exert its effect during diet-induced obesity, and several molecular alterations have been associated with attenuated leptin signalling, inducing the metabolic syndrome parameters reduced by a PBJ diet. These results well agree with an intervention study in obese patients, showing that bergamot polyphenol extract complexed with pectin decreased body weight by 14.8% and body mass index by 15.9% in a treated group with respect to a control group. The body weight effect correlated with a significant reduction of circulating hormones balancing caloric intake, including leptin, ghrelin and upregulation of adiponectin [14].

## 4. Materials and Methods

### 4.1. Chemicals

The trolox (6-Hydroxy-2,5,7,8-tetramethyl-3,4-dihydrochromene-2-carboxylic acid), gallic acid, naringenin 7-glucoside, luteolin, stachydrine, 2,2-diphenyl-1-picrylhydrazyl (DPPH), sodium acetate, acetic acid, ethanol, methanol, formic acid and LC-MS grade solvents were from Merck KGaA, Darmstadt, Germany. A Milli-Q H_2_O purification system (Millipore, Bedford, MA, USA) was used for the LC-grade H_2_O (18 MΩ cm) preparation.

### 4.2. PBJ Preparation

Bergamot (*Citrus Bergamia* Risso and Poiteau) fruits (100 kg) of three cultivars “Castagnaro”, “Femminello” and “Fantastico” (equally represented) grown in the coastal area of Calabria (southern Italy), from Reggio Calabria to Monasterace, were harvested from November to February. The fruits were washed in steel tanks with water; once peeled, the fruits were pressed in a stainless steel twin-screw press, yielding juice (fruit juice) and solid residue (pulp, also named pastazzo). The pulp was then backwashed three times with deionized water with a pulp:deionized water ratio of 1.5:1 to obtain, after a depulping process with a decanter and centrifuges, a juice (pulp juice or technical juice), which was mixed with fruit juice. Pulp washing was carried out in a horizontal prismatic tank made of stainless steel with a stainless steel agitator. Liquid–solid separation was performed using a stainless steel rotary filter. The resulting juice (fruit + technical juice) was then added with gum arabic (20%) as a drying aid with the purpose of reducing particle adhesion, hygroscopicity and the degree of caking, and for increasing the stability of the powder during storage [32]. After adjusting the pH to 4.5 with KOH, the juice was dried in a spray dryer to give 2.8 Kg of solid dry juice (powder of bergamot juice, PBJ; yield of 2.8 with respect to the starting material). The spray drying of the juice was carried out using a semi-industrial spray dryer Niro Atomizer type FU 11 DA (Søbork, Denmark) with a diameter of 1.2 m, and the rotary atomizer operation was set at a speed of 12,000 rpm. Drying was carried out at inlet and outlet temperatures of 150 and 80 °C, respectively. the flow rate of the feed solution was in the range of 10–15 L/h as the control variable and the drying air flow rate was maintained at 460 m^3^/h. Based on the promising in vitro and preclinical studies, the PBJ production was then scaled at an industrial level by AKHYNEX srl, (Polistena, RC, Italy) and will be commercially available with the tradename of Endo@berg by AKHYNEX and KALITA^®^ by Giellepi S.p.A. (Milan, Italy).

### 4.3. Analytical Characterization and Nutritional Values

#### 4.3.1. Polyphenols Profiling by LC-ESI-MS

The PBJ sample was dissolved in a CH_3_OH/H_2_O mixture (50:50, % *v*/*v*) and diluted 1:2 in H_2_O/HCOOH, 100/0.1 % *v*/*v* (mobile phase A) to obtain the final concentration of 3 mg/mL. Trolox 5 × 10^−5^ M was added to the sample as an internal standard. The analysis was performed in triplicate by LC-HRMS by applying the method described by Baron et al. [20], acquiring MS spectra in both positive and negative ion modes. Xcalibur 4.0 software was used to analyze the spectra. The data were evaluated by a targeted approach based on the database of compounds reported in the paper by Baron et al. [3] with the addition of osmolyte components reported by Slama et al. [33]. The putative identification of the compounds is based on the assessment of the accurate mass, isotope and fragmentation pattern and retention time, according to our previous paper [3].

#### 4.3.2. Quantitative Analysis of Polyphenols by HPLC-PDA

The total polyphenol content of PBJ was determined by using HPLC coupled with a photo-diode-array (PDA) detector and an external standard method. In more detail, the PBJ solutions were prepared in CH_3_OH/H_2_O 50:50 % *v*/*v* and diluted for analysis to a final concentration of 1 mg/mL using H_2_O/HCOOH, 100/0.1, % *v*/*v* (mobile phase A). A volume of 10 µL was analyzed in triplicate using an HPLC system (Surveyor, Thermo Finnigan Italy, Milan, Italy) equipped with a PDA analyser (Surveyor, Thermo Finnigan Italy, Milan, Italy) and an Agilent Zorbax SB-C18 reverse phase column (150 × 2.1 mm, i.d. 3.5 μm, CPS analitica, Milan, Italy). An eighty-min multistep gradient of mobile phases A and B (CH_3_CN/HCOOH, 100/0.1, % *v*/*v*) was used for the separation of the polyphenols present in PBJ as follows: the % of mobile phase A was reduced from 90% to 80% from 0 to 45 min, to 40% from 45 min to 65 min and to 10% within 1 min and kept at this value until 70 min. The % of phase A was then returned to 90% within 1 min and equilibrated for 9 min before the next analysis.

The flow rate was 200 µL/min and the column was kept at 40 °C while the PDA detector was set to monitor the absorbance from 200 to 600 nm. For quantification, three calibration curves were built using one standard for each subclass of polyphenol contained in the PBJ extract: gallic acid for phenolic acids (λmax 271; 5–50 µg/mL), naringenin 7-glucoside for flavanones (λ_max_ 283; 15–100 µg/mL) and luteolin flavones (λmax 348; 15–100 µg/mL). Each identified polyphenol was assigned to the corresponding polyphenol subclass on the basis of the absorption wavelength and the concentration calculated using the calibration curve carried out for the corresponding polyphenol subclass. The total amount of polyphenols was calculated by the sum of the concentration of each identified polyphenol and the results are reported as a *w*/*w* percentage.

#### 4.3.3. Stachydrine Content by LC-HRMS Standard Addition Method

The stachydrine in PBJ was determined by HPLC-MS using the standard addition method to avoid the matrix effect. The PBJ samples were prepared in a CH_3_OH/H_2_O mixture (50:50, % *v*/*v*) and diluted in H_2_O/HCOOH, 100/0.1 % *v*/*v* (mobile phase A) to obtain a final extract concentration of 50 µg/mL. The samples with a fixed extract concentration were then spiked with increasing amounts of standard stachydrine to obtain the following final concentrations of added stachydrine: 0 (no stachydrine spiking), 0.5, 1.0, 1.5 and 2 µM. The analysis was performed in triplicate by applying a multistep LC-HRMS method (H_2_O/HCOOH, 100/0.1 % *v*/*v* mobile phase A; CH_3_CN/HCOOH, 100/0.1 % *v*/*v* mobile phase B; 10 min gradient: 0 min 1% B; 5 min 90% B; 6.9 min 90% B; 7 min 1% B; 10 min 1% B), using the Orbitrap analyzer in full scan positive ion mode. The chromatographic method involved the use of an Agilent Zorbax SB-C18 column, 3.5 µm, 2.1 × 150 mm, maintained at 40 °C with a flow rate of 300 µL/min and an injected volume of 20 µL. Xcalibur 4.0 software was used to analyze the spectra. A calibration curve was then generated by integrating the stachydrine chromatographic peaks and placing the known added stachydrine concentrations (µM) on the *x*-axis and the AUC of the corresponding chromatographic peak on the *y*-axis; the equation of the straight line was derived using the linear regression method and the stachydrine content in the PBJ was calculated by setting y = 0. The results are expressed as % *w*/*w* of the dry extract.

#### 4.3.4. Semi-Quantitative Data Analysis

A semi-quantitative analysis of each metabolite identified by LC-MS was carried out by reconstituting the corresponding single ion chromatogram (SIC) by setting the molecular ion as filter ion and a tolerance of 5 ppm. The area under the curve of the peak relative to each metabolite was automatically integrated as that of the internal standard (Trolox). The ratio between the AUC of each metabolite (AUCn) and the AUC of the IS (AUC_Trolox_) was then calculated and divided by the sum of the ratios of all the compounds and expressed as % as reported by the Equation (1).
Relative abundance (%) = (AUC_n_/AUC_Trolox_)/∑(AUC_n_/AUC_Trolox_)(1)

#### 4.3.5. Nutritional Composition

The proximate composition of PBJ was determined using the Association of Official Analytical Chemists (AOAC, http://www.aoacofficialmethod.org, accessed on 16 January 2023) and International Organization for Standardization (ISO, https://www.iso.org/home.html, accessed on 18 January 2023) methods. In particular, the following components were determined: moisture (AOAC 934.06-1934(1996), loss on drying (moisture) in dried fruits), proteins (AOAC 954.01-1954(1996), protein (crude) in animal feed and pet food; Kjeldahl method), total carbohydrates (AOAC 971.18-1980, carbohydrates in fruit juices; gas chromatographic method), total fat (AOAC 920.39-1920, fat (crude) or ether extract in animal feed), ash (AOAC 940.26-1940, ash of fruits and fruit products), soluble and insoluble fibers (AOAC AOAC 991.43-1994(2000), soluble and insoluble dietary fiber in foods; enzymatic-gravimetric method, MES-TRIS buffer), fructooligosaccharide (ISO 22579:2020, IDF 241:2020, infant formula and adult nutritionals—determination of fructans; high performance anion exchange chromatography with pulsed amperometric detection (HPAEC-PAD) after enzymatic treatment) and sodium (AOAC 966.16-1968, sodium in fruits and fruit products; flame spectrophotometric method). The energy value was calculated according to the Atwater extended system [34]. The fatty acids were determined by GC-FID (Agilent 7890B/FID, Agilent Technologies, Santa Clara, CA, USA) following the AOAC 2012.13 method. The sugar composition was assessed by HPLC coupled with a pulsed amperometric detection (PAD) (Thermo Scientific Dionex ICS-5000, Sunnyvale, CA, USA) as described by Zhuoi et al. [35], focusing on glucose, fructose, lactose, saccharose and maltose (Merck KGaA, Darmstadt, Germany). The organic acids (malic acid, succinic acid, lactic acid, fumaric acid, tartaric acid, citric acid, formic acid, propionic acid and acetic acid) were quantified by HPLC-PDA according to Schrerer R. et al. [36] using an HPLC system equipped with a PDA analyser (Surveyor, Thermo Finnigan Italy, Milan, Italy) and an Agilent Zorbax SB-C18 reverse phase column (150 × 2.1 mm, i.d. 3.5 μm, CPS analitica, Milan, Italy).

### 4.4. In Vitro and Cell Activities

#### 4.4.1. In Vitro Radical-Scavenging Activity

The antioxidant activity was assessed by the DPPH assay. A 50:50 (% *v*/*v*) H_2_O:EtOH solution of the PBJ extract was prepared and diluted to obtain the final extract concentrations in the range of 50–1000 µg/mL. The reaction mixture was prepared by mixing aliquots of 500 µL of the extract, 1 mL of acetate buffer (100 mM, pH 5.5), 1 mL of ethanol and 500 µL of an ethanolic DPPH solution (500 µM). After an incubation of 90 min in the dark, the absorbance was read at 517 nm using a Shimadzu UV 1900 spectrophotometer (Shimadzu, Milan, Italy). The results are expressed as a mean of the percentage of inhibition ± SD (Table 3); the IC_50_ values obtained were compared with those of trolox and ascorbic acid, which were chosen as reference compounds for their antioxidant activity.

#### 4.4.2. Anti-Inflammatory Activity

The anti-inflammatory activities of PBJ and stachydrine were tested using the previous in vitro cell model described by Baron et al. [3]. The PBJ was tested as such after a treatment to remove the fibrous and insoluble fraction, so as to free the polyphenol fraction embedded in the matrix to make it more available to the cells. The treatment consisted of sonicating PBJ in a hydroalcoholic mixture (CH_3_CH_2_OH/H_2_O, 70/30 % *v*/*v*) at a ratio of 1:10 *m*/*v* for 30 min and centrifuging the mixture for 5 min at 10,000 rpm. The supernatant was then separated from the fibrous pellet and dried with a centrifugal evaporator at 30 °C for 18 h. The percentage extraction yield was of 77.922 ± 1.181%, and the qualitative composition of the polyphenols was maintained.

For the anti-inflammatory cell test, the R3/1 NF-kb cell line [3] was seeded in a 96-well white plate (BRANDplates^®^, cell grade) at a density of 4000 cells/well. The cells were pre-treated with different concentrations of the untreated and extracted/sonicated PBJ (1 µg/mL–250 µg/mL) and standard stachydrine (1–200 µM) for 18 h in complete medium (DMEM, 10% FBS, 1% L-glutamine, 1% penicillin/streptomycin). Subsequently, an inflammatory state was induced for 6 h using 10 ng/mL TNFα. To evaluate the luciferase activity, the cells were washed once with 100 µL of 1× PBS, and finally 50 µL of DMEM was added to each well. An aliquot of 50 µL of ONE-Glo™ Luciferase Assay Substrate (purchased from Promega Corporation, Madison, WI, USA) was added directly to the cells, and then the luciferase was measured using a luminometer (Wallac Victor2 1420, Perkin-Elmer™ Life Science, Monza, Italy). The experiments were performed with technical and biological replicates. The cell viability was tested at all the concentrations used in the anti-inflammatory assay by means of the MTT assay performed on the same cell line and under the same conditions.

#### 4.4.3. Antioxidant Activity

For the evaluation of antioxidant activity, the NRF2/ARE Responsive Luciferase cell line HEK293 was used (Signosis, Santa Clara, CA, USA). The cells were seeded in white 96-well plates (BRANDplates^®^, cell grade) with a density of 10,000 cells/well and subsequently treated with the PBJ as such, or after hydroalcoholic extraction under sonication, as previously described. The PBJ and stachydrine were added to complete medium (DMEM, 10% FBS, 1% L-glutamine, 1% penicillin/streptomycin) in a concentration range of 1–250 µg/mL for 18 h. To avoid reading interference, the medium was removed and 50 µL of PBS were added to each well. Aliquots of 50 µL of ONE-Glo™ Luciferase Assay Substrate (purchased from Promega Corporation, Madison, WI, USA) were added directly to the cells, and then the luciferase was measured using a luminometer (Wallac Victor2 1420, Perkin-Elmer™ Life Science, Monza, Italy). The experiments were performed with technical and biological replicates.

### 4.5. Pharmacological Studies

#### 4.5.1. Experimental Protocol and Group Characterization

The experiment was approved by the Animal Ethics Committee of Botucatu Medical School, São Paulo State University (1337/2020), and was performed in accordance with the National Institute of Health’s Guide for the Care and Use of Laboratory Animals. Male *Wistar* rats (±160 g) were kept in a controlled environment at a temperature of 22 °C ± 3 °C, luminosity with a 12 h light–dark cycle, relative humidity of 60 ± 5% and randomly distributed into four experimental groups to receive: control diet + placebo (C, *n* = 16); control diet + PBJ (C + PBJ, *n* = 15); high-sugar, high-fat diet + placebo (HSF, *n* = 9); or high-sugar, high-fat diet + PBJ (HSF + PBJ, *n* = 9) over a 20 week period. The HSF diet groups also received water + sucrose (25%). At the end of the 20 weeks, the animals were fasted for 8 h, anesthetized (thiopental 120 mg/kg/i.p.) and euthanized. Blood samples were collected and plasma was obtained. The adipose tissue was dissected and weighed for nutritional parameter assessment. Food and water were offered ad libitum and the diets were designed in our laboratory as previously published [37].

#### 4.5.2. PBJ Administration

The administration of PBJ was performed by daily gavage at a concentration of 250 mg/kg. Drinking water was used as a dilution vehicle for the extract; the animals in the placebo groups only received the drinking water by gavage daily in the same way as the treated group.

#### 4.5.3. Nutritional Parameters

The nutritional profile was evaluated according to chow feed, water intake, caloric intake and adiposity index. The chow and water intake also considered the animals’ daily leftovers. The caloric intake was determined by multiplying the energy value of each diet (g × Kcal) by the daily food consumption. For the HSF group, the caloric intake also considered the calories from water with sucrose (0.25 × 4 × consumed volume ). The adiposity index was used as an indicator of obesity since it accurately assesses the amount of body fat in the animals. After euthanasia, the epididymal, visceral and retroperitoneal fat deposits were dissected from the animals and the sum of the deposits normalized by body weight [(epididymal + retroperitoneal + visceral)/body weight x 100] is considered the adiposity index [37,38].

#### 4.5.4. Metabolic Analysis

After 8-h of fasting, blood was collected, and the plasma was used to measure biochemical parameters. The glucose concentration was determined in a blood drop using a glucometer (Accu-Chek Performa; Roche Diagnostics Brazil Limited, São Paulo, Brazil); the triglycerides levels were measured with an automatic enzymatic analyzer system (Chemistry Analyzer BS-200, Mindray Medical International Limited, Shenzhen, China) [39]. The insulin resistance was estimated according to the fasting triglycerides and glucose index (TyG) using the following formula [40]:Ln [fasting triglycerides (mg/dL) × fasting glucose (mg/dL)]/2

#### 4.5.5. Systolic Blood Pressure

Systolic blood pressure (SBP) evaluation was assessed in conscious rats by the non-invasive tail-cuff method with a NarcoBioSystems^®^ Electro-Sphygmomanometer (International Biomedical, Austin, TX, USA). The animals were kept in a wooden box (50 cm × 40 cm × 30 cm) between 38 and 40 °C for 4–5 min to stimulate arterial vasodilation [41]. After this procedure, a cuff with a pneumatic pulse sensor was attached to the tail of each animal. The cuff was inflated to 200 mmHg pressure and subsequently deflated. The blood pressure values were recorded on a Gould RS 3200 polygraph (Gould Instrumental, Valley View, OH, USA). The average of three pressure readings was recorded for each animal.

#### 4.5.6. Statistical Analysis

The data are presented as means ± standard deviation (SD) or medians (interquartile range). For the in vitro and cell experiments, a statistical analysis was performed on the results obtained using a one-way ANOVA with Bonferroni’s multiple comparison test (*p* < 0.05 was considered significant). For the in vivo studies, the differences among the groups were determined by two-way analysis of variance. Statistically significant variables were subjected to the two-way ANOVA test with Tukey post-hoc. Statistical analyses were performed using Sigma Stat for Windows Version 3.5. (Systat Software, Inc., San Jose, CA, USA). A *p* value of 0.05 was considered as statistically significant.

## 5. Conclusions

In conclusion, in the present paper, we have fully characterized a sustainable industrial raw material derived from the bergamot industrial chain containing multiple bioactive components and, in particular, fibers, polyphenols and non-polyphenolic small molecules, such as stachydrine. The polyphenols and stachydrine were found to act as anti-inflammatory components through different mechanisms of action, but not involving the Nrf2 pathway. PBJ was then found to be effective in an in vivo model of metabolic syndrome and, in addition to considering the well-known molecular effects already reported for bergamot polyphenols (anti-inflammatory, antioxidant and lipid regulating effect), additional biological activities should be noted in the dietary fibers and to the non-phenolic constituents, such as stachydrine. Among the molecular mechanisms explaining the in vivo effect on metabolic syndrome, the effect of PBJ constituents on the regulation of energy homeostasis through leptin networking should also be considered, and since this was not explored in the present paper, it deserves further investigation.

In conclusion, this paper reports a full nutritional and chemical characterization, together with the health benefit properties of a non-purified powdered bergamot juice. Since bergamot juice extract is often considered a by-product of essential oil production, whose disposal poses a significant economic and environmental challenge, these data contribute to identifying bergamot juice as a source of healthy ingredients for the preparation of nutraceutical health products.

We believe that the data reported here will stimulate the scientific community to further investigate bergamot juice as a valuable source of bioactive components, and will also encourage various industries to seek potential uses of bergamot juice and derivatives in commercial products.

## Figures and Tables

**Figure 1 molecules-28-02964-f001:**
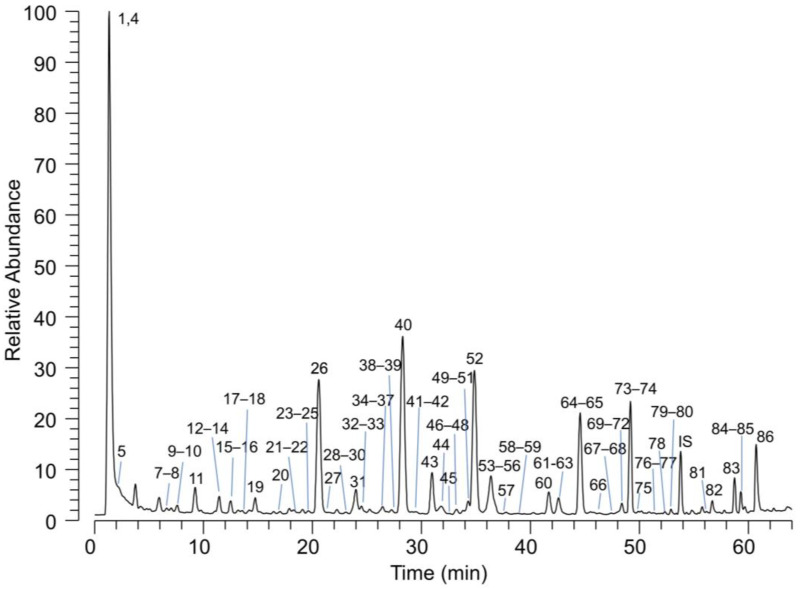
LC-MS total ion current chromatogram (TIC) of PBJ acquired in negative ion mode. The numbers refer to the identified compounds listed in Table 2.

**Figure 2 molecules-28-02964-f002:**
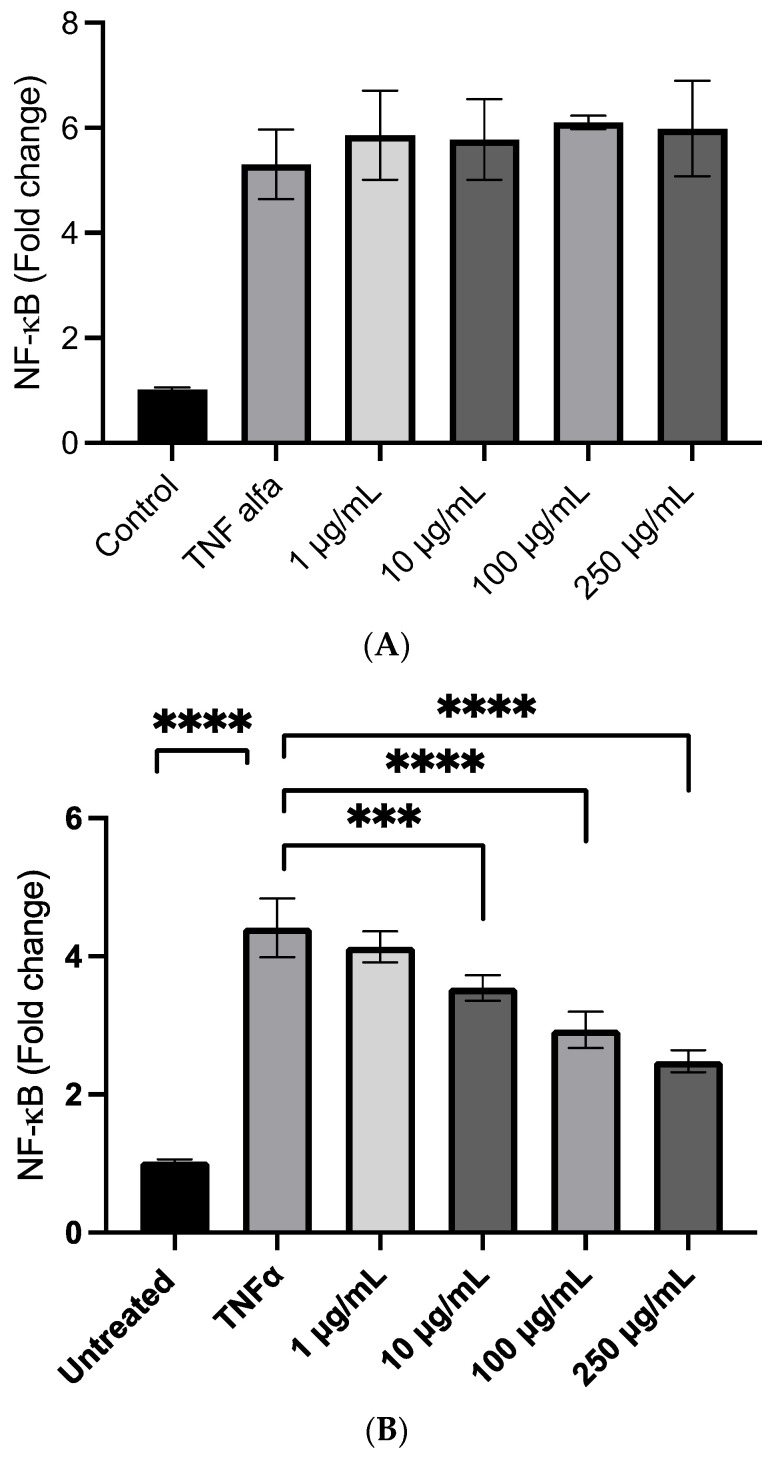
The anti-inflammatory activity of PBJ and stachydrine in the R3/1 cell line with a gene reporter for NF-kb. The effects of PBJ are shown before (**A**) and after (**B**) extraction in a hydro-alcoholic mixture and under sonication to release the small molecules from the polymeric matrix. The dose-dependent anti-inflammatory effect of stachydrine is shown in panel (**C**). *** *p* < 0.001; **** *p* < 0.0001.

**Figure 3 molecules-28-02964-f003:**
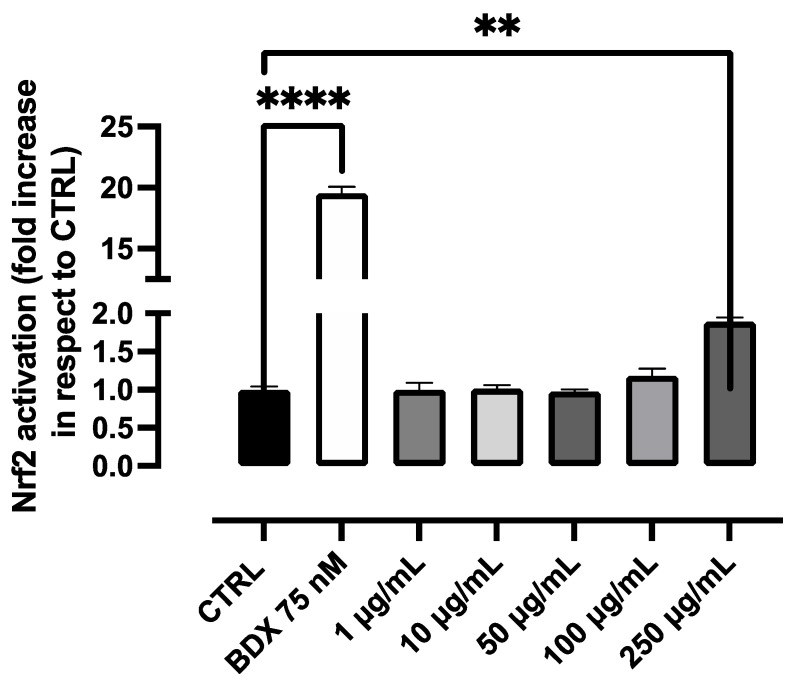
Antioxidant activity of PBJ in the R3/1 cell line with a gene reporter for NRF2. ** *p* < 0.01; **** *p* < 0.0001.

**Figure 4 molecules-28-02964-f004:**
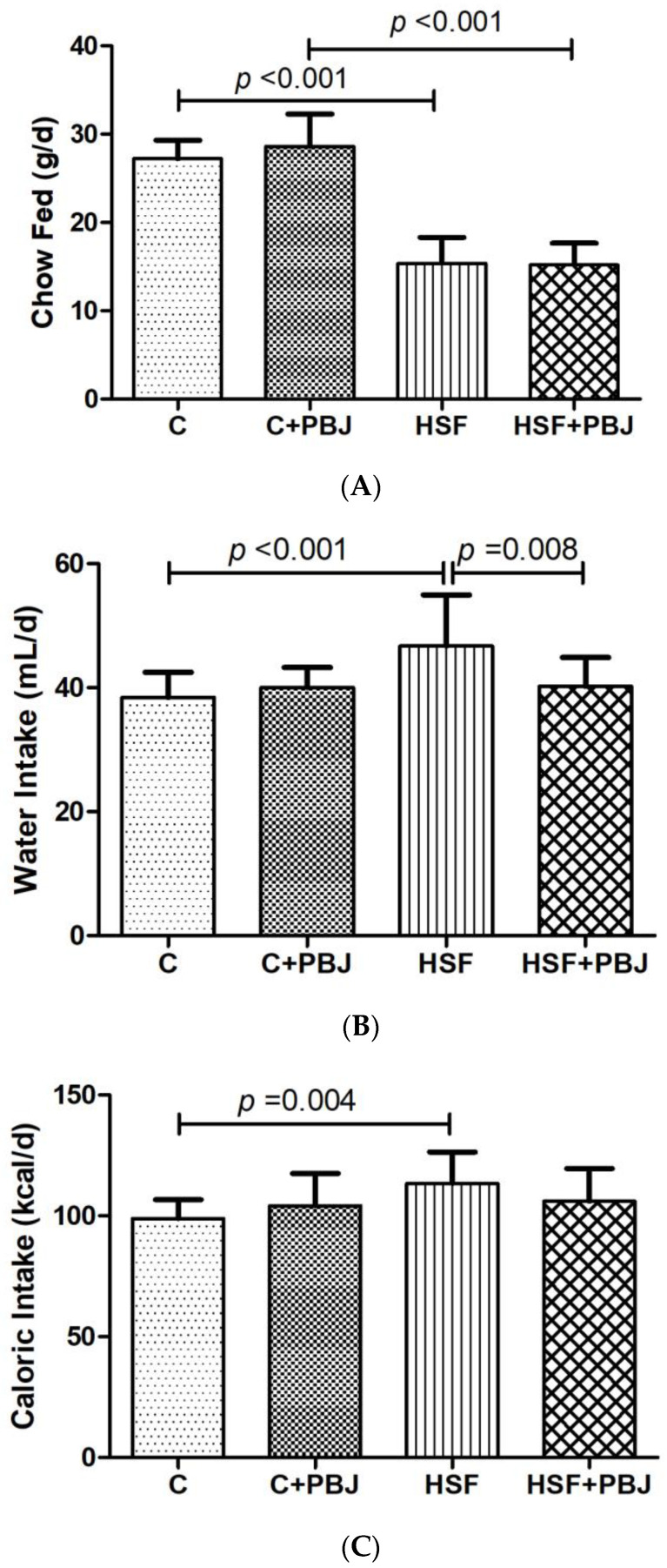
Nutritional Intake. (**A**) Chow fed (g/d); (**B**) Water intake (mL/d); (**C**) Caloric intake (kcal/d). The data are expressed as mean ± standard deviation (SD). Comparison by two-way ANOVA with Tukey’s post-hoc. C—Control diet + placebo; HSF—High-sugar, high-fat diet; *p* < 0.05 as significant.

**Figure 5 molecules-28-02964-f005:**
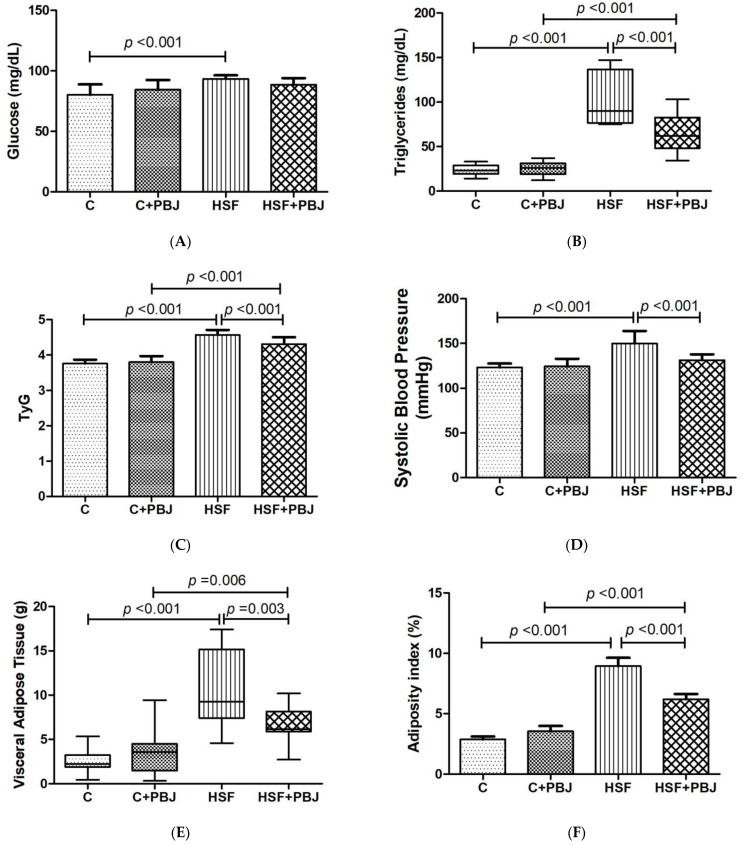
Metabolic syndrome parameters. (**A**) Glucose (mg/dL); (**B**) Triglycerides (mg/dL); (**C**) TyG; (**D**) Systolic blood pressure (mmHg); (**E**) Visceral adipose tissue (g); (**F**) Adiposity index (%); (**G**) Final body weight. The data are expressed as mean ± standard deviation (SD) or medians (interquartile range). Comparison by two-way ANOVA with Tukey’s post-hoc. TyG—Fasting triglycerides and glucose index. C—Control diet + placebo; HSF–High-sugar, high-fat diet; *p* < 0.05 as significant.

**Table 1 molecules-28-02964-t001:** PBJ nutrients composition and energy values.

Nutritional Components		Value
Moisture		11.68 ± 0.38 g/100 g
Proteins		5.65 ± 0.35 g/100 g
Total carbohydrates		57.250 ± 2.088 g/100 g
Total fat		0.240 ± 0.036 g/100 g
Ash		5.78 ± 0.30 g/100 g
Energy value (kcal/100 g)		316 ± 3 kcal/100 g
Energy value (kJ/100 g)		1348 ± 11 kJ/100 g
Fatty acids		
	Polyunsaturated (>C_20_)	<LOQ (0.0010 g/100 g)
	Saturated	0.079 ± 0.014 g/100 g
	Monounsaturated	0.0210 ± 0.0070
	Polyunsaturated	0.127 ± 0.021
Total dietary fibers		19.4 ± 2.0 g/100 g
	Dietary soluble fibers	18.2 ± 1.9 g/100 g
	Dietary insoluble fibers	1.20 ± 0.35 g/100 g
Fructooligosaccharide		<LOQ (0.10 g/100 g)
Sodium		1370 ± 110 mg/Kg
Sugar composition		
	Glucose	12.33 ± 0.90
	Fructose	10.57 ± 0.68
	Lactose	<LOQ
	Saccharose	3.09 ± 0.30
	Maltose	<LOQ
Organic acids		
	Malic acid	1.21 ± 0.015 g/100 g
	Succinic acid	<LOQ (0.010 g/100 g)
	Lactic acid	<LOQ (0.010 g/100 g)
	Fumaric acid	<LOQ (0.010 g/100 g)
	Tartaric acid	<LOQ (0.010 g/100 g)
	Citric acid	13.8 ± 1.6 g/100 g
	Formic acid	<LOQ (0.010 g/100 g)
	Propionic acid	<LOQ (0.010 g/100 g)
	Acetic acid	<LOQ (0.010 g/100 g)

**Table 2 molecules-28-02964-t002:** Qualitative and semi-quantitative composition of small molecules in PBJ identified by HPLC coupled with a high-resolution MS.

Peak	Compound	RT	[M-H]-	MS/MS	[M+H]+	MS/MS	Ionic Relative %
1	Quinic acid	1.3	191.0564	147	-	-	1.7
2	Betonicine	1.4	-	-	160.0967	88-98-102-114-160	<1.0
3	Stachydrine	1.4	-	-	144,1019	84-144	<1.0
4	Citric acid	1.5	191.0202	111-147	193.0349	129-147-157	7.5
5	HMG-glucoside	1.9	323.0974	-	-	-	<1.0
6	N-(1-Deoxy-1-fructosyl)leucine	2	-	-	294.1549	144-230-248-258	<1.0
7	Luteolin-6,8-di-C-glucoside	6.7	609.1443	369-399-429-471-489-519	611.1612	371-473	<1.0
8	Feruloyl glucoside isomer 1	7.1	355.1029	193	357.1182	287-195	<1.0
9	Sinapoyl glucoside	7.8	385.1133	223	387.1288	225	<1.0
10	Citrusin F	8.6	519.1714	195-357	*-*	-	<1.0
11	Apigenin-6,8-di-C-glucoside	9.4	593.1495	353-383-473-503	595.1657	355-457	2.3
12	Chrysoeriol-6,8-di-C-glucoside	11.3	623.1612	312-383-413-503-533	625.1762	385-487-505	<1.0
13	2-Hydroxy-4-methoxyhydrocinnamoyl-2-O-glucoside	11.6	357.1179	151-177-195	-	-	<1.0
14	*p*-Coumaric acid	11.8	163.0406	119	-	-	<1.0
15	Diosmetin-6,8-di-C-glucoside	12.9	623.1604	312-383-413-503-533	625.1761	385-487-505	1.4
16	Neoeriocitrin-O-glucoside/eriocitrin-O-glucoside	13.2	757.2193	287-595	759.2349	-	<1.0
17	Luteolin-C-glucoside	13.6	447.0924	285	449.1079	287-329	<1.0
18	Naringin-glucoside	13.9	741.2249	271	-	-	<1.0
19	Unknown 1	14.5	611.1617	449-475	613.1762	-	<1.0
20	6-(beta-D-glucopyranosyloxy)-5-benzofuranpropanoic acid	16.9	367.1028	161-205	-	-	<1.0
21	Eriodictyol-7-O-glucoside	18.2	449.108	287	451.1235	289	<1.0
22	Apigenin-8-C-glucoside	18.3	431.0983	283-311-341	433.1131	283-397	<1.0
23	Eriodictyol 7-O-neohesperidoside (Neoeriocitrin)	18.8	595.1651	287-449	597.1813	451	<1.0
24	Apigenin-6-C-glucoside	19.1	431.0979	283-311-341	433.113	283-313-337-367-379-397	<1.0
25	Apigenin-di-O-glucoside-O-HMG	19.3	737.1903	431-635-675	-	-	<1.0
26	Eriodictyol 7-O-rutinoside (Eriocitrin)	20.6	595.1657	287	597.1812	289-451	11.7
27	6-(beta-D-glucopyranosyloxy)-4-methoxy-5-benzofuranpropanoic acid	20.7	397.1135	176-191-217-235	399.1288	202-219-237-245-263-285-288-313-325-339	<1.0
28	Chrysoeriol-8-C-glucoside	22.7	461.1084	-	463.1234	325-343-367-381-397-409-427-445	<1.0
29	Eriodictyol-O-diglucoside-O-HMG	23	755.2003	287-449-491	-	-	<1.0
30	Diosmetin-di-C-glucoside-O-HMG	23.7	767.1998	299-461	-	-	<1.0
31	Luteolin-7-O-neohesperidoside	24.1	593.1496	285-447	595.1653	287-449	2.5
32	Diosmetin-8-C-glucoside	24.5	461.109	341-371	463.1228	343-367-397-409-427	<1.0
33	Naringenin 7-O-rutinoside (Narirutin)	25.2	579.1713	271	581.1863	435	<1.0
34	Naringenin-7-O-glucoside (Prunasin)	26.5	433.1133	271	-	-	<1.0
35	Neoeriocitrin-glucoside-O-HMG	26.5	901.2591	287-595-637-677-799	-	-	<1.0
36	Hesperetin-di-C-glucoside	26.8	625.1776	301-343-505	627.1924	303	<1.0
37	Chrysoeriol-di-O-glucoside-O-HMG	26.9	767.1998	461	-	-	<1.0
38	Diosmetin-di-O-glucoside-O-HMG	28	767.2006	-	-	-	<1.0
39	Bergamjuicin (Melitidin-glucoside)	28	885.264	459-579-621-661-723-741-783	887.282	273	<1.0
40	Naringenin 7-O-neohesperidoside (Naringin)	28.4	579.1705	271	581.1863	417-435	11.4
41	Apigenin-O-glucoside	29.1	431.0976	-	433.1132	-	<1.0
42	Neohesperidin-glucoside-O-HMG	29.2	915.273	-	-	-	<1.0
43	Apigenin-7-O-neohesperidoside	31	577.1557	269	579.1705	271-433	2.3
44	Hesperetin 7-O-rutinoside (Hesperidin)	31.4	609.1816	301-489	611.1971	-	<1.0
45	Chrysoeriol-7-O-glucoside	32.4	461.1079	-	463.1232	301	<1.0
46	Diosmetin-7-O-glucoside	33.2	461.1082	284-299	463.1233	301	<1.0
47	Eriodictyol 7-O-glucoside-O-HMG	33.5	593.1495	287-449-491-531	595.1654	-	<1.0
48	Hesperetin-O-glucoside isomer 1	33.7	463.1239	301	-	-	<1.0
49	Eriocitrin-O-HMG	34.1	739.2052	287-595-637-677	741.2234	-	<1.0
50	Chrysoeriol-7-O-neohesperidoside	34.3	607.1655	284-299-461	-	-	<1.0
51	Demethoxycentaureidin-7-O-glucoside	35	491.1199	314-329	493.1339	331	<1.0
52	Hesperetin 7-O-neohesperidoside (Neohesperidin)	35	609.1809	301-489	611.1965	303	16.5
53	Diosmetin-7-O-neohesperidoside	36.2	607.1655	284-299	-	-	1.8
54	Hesperetin-O-glucoside isomer 2	36.5	463.1241	-	-	-	<1.0
55	Neoeriocitrin-O-HMG	36.5	739.2067	595-637-677	741.2233	-	3.8
56	6-(beta-D-glucopyranosyloxy)-4-methoxy-5-benzofuranpropanoic acid-O-HMG	36.7	541.1556	191-217-235-397-439-479	543.1704	325-367-499	1.1
57	6-hydroxy-4-methoxy-5-benzofuranpropanoic acid	38	235.0611	176-191	-	-	<1.0
58	Luteolin-7-O-neohesperidoside-O-HMG	38.5	737.1934	593-635-675	739.2083	593-677	<1.0
59	Eriodictyol	39.2	287.056	-	-	-	<1.0
60	Nomilin glucoside	41.7	693.2748	427-471-565-607-633	-	-	3.2
61	Naringenin-acetyl-C-neohesperidoside	42.6	621.1812	271-459-501-579	-	-	<1.0
62	Naringenin-O-rutinoside-O-HMG	42.7	723.2142	579-621-661	725.2291	-	<1.0
63	Naringenin 7-O-glucoside-O-HMG	43	577.1554	271-433	579.1711	-	<1.0
64	Nomilinic acid glucoside	44.6	711.285	607-651	-	-	4.1
65	Melitidin (Naringin-O-HMG)	44.6	723.2148	579-621-661	725.2289	689	6.7
66	Apigenin-7-O-neohesperidoside-O-HMG	46.3	721.1959	577-619-659	723.2133	271	<1.0
67	Chrysoeriol-O-glucoside-O-HMG	47.4	605.1508	299-461-503	607.1652	301-463	<1.0
68	Diosmetin-acetyl-O-neohesperidoside	47.8	649.1771	284-299-607	-	-	<1.0
69	Obacunone glucoside	48.3	633.2535	331-359-427	-	-	<1.0
70	Diosmetin-7-O-neohesperidoside-O-HMG	48.3	751.2095	607-649-689	753.2227	607-691	<1.0
71	Hesperetin-O-rutinoside-O-HMG	48.4	753.2223	609-651-691	755.2392	301-609	1.4
72	Hesperetin 7-O-glucoside-O-HMG	48.8	607.1651	301-463-505	-		<1.0
73	Brutieridin (Neohesperidin-O-HMG)	49.2	753.2221	301-609-651-691	755.2389	301-609	6.9
74	Diosmetin-O-glucoside-O-HMG	49.3	605.1505	299-461-503-543	607.1655	301	<1.0
75	Demethoxycentaureidin-7-O-glucoside-HMG	49.8	635.1604	329-491-533	637.1761	331	<1.0
76	Unknown 2	51.3	417.0819	129-173-251-295	-	-	<1.0
77	Naringenin	51.5	271.0608	107-151-177	-	-	<1.0
78	Unknown 3	52.5	417.0816	129-173-251-295	-	-	<1.0
79	255-C-glucoside-O-rhamnoside-O-HMG	53.1	707.2159	255-357-563-605-645	709.2337	-	<1.0
80	Isosakuranetin-7-O-neohesperidoside-O-HMG	53.4	737.2266	285-593-635-675	739.244	-	<1.0
81	Deacetylnomilinic acid	56.1	489.2125	325-333-411	491.2275	341-369-385-411-455	<1.0
82	Limonoate A-ring lactone	56.7	487.1953	383-427	489.2119	369-383-427-445	<1.0
83	Limonin	58.7	469.1872	229-278-283-306-321-381	471.2012	367-409-425	<1.0
84	Obacunoic acid	59.4	471.2017	307-325-409	473.2172	-	<1.0
85	Nomilinic acid	59.4	531.222	427-471-489	533.2383	341-369-411	<1.0
86	Deacetylnomilin	60.2	471.2015	307-325-409	473.2173	-	<1.0

**Table 3 molecules-28-02964-t003:** Radical scavenging activity of PBJ.

Sample	IC_50_ (µg/mL)
PBJ	233.6 ± 18.0 (30 µg/mL expressed as polyphenols)
Trolox	5.0 ± 0.3
Ascorbic acid	3.918 ± 0.047

## Data Availability

Data are contained within the article.

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
