# Peer review of "Chemical, Nutritional and Biological Evaluation of a Sustainable and Scalable Complex of Phytochemicals from Bergamot By-Products"

_molecules, 2023, doi:10.3390/molecules28072964_

Round 1

Reviewer 1 Report

Manuscript entitled:” Chemical, nutritional and biological evaluation of a sustainable and scalable complex of phytochemicals from bergamot by-products

The manuscript describes a procedure for preparation of powdered bergamot juice (PBJ), and characterized the chemical and nutritional properties. In the second part of the manuscript, the authors evaluated the biological activity of PBJ in cell models to test its anti-inflammatory and antioxidant activity and in an in vivo rat model of metabolic syndrome induced by a high caloric intake.

In my opinion, the manuscript needs several improvements especially in Materials and methods part.

Minor comments:

1.     The term in vivo, in vitro and scientific names should be written in Italic form through all the manuscript.

2.     Some sentences showed be revised/rewritten again to avoid repeated words or grammatical English language mistakes. For example, Page 3 line 5, 15, 25; Page 3 last paragraph line 3 (built correct to carried out); Page 4 subtitle 2.3.4  line 4 (integrated as was that…..correct to integrated as that)…………….and others

Major comments:

1.     The manuscript was not prepared carefully according to the journal guidelines. Where the orders of the main parts are not correct. It should be Introduction, Results and discussion, Materials and methods, Conclusion and references.

2.     Materials and methods part showed be enhanced by adding some details about:

a.      Nutritional composition, which components were determined? and give the complete citation for AOAC (Year) with methods numbers and also for ISO. The AOAC and ISO standard methods should be added in References list.

b.     The authors should give the reason for Arabic gum addition in this stage (Page 3 line 7).

c.      Page 5, subtitles 2.3.4 and 2.4.3. regarding the statistical analysis for anti-inflammatory and antioxidants; it should be removed from these subtitles then added in part 2.5.6. Statistical analysis.

3.     Results need some improvements

a.      Table (1), there are no details were mentioned in Materials and Methods part about the fatty acids, dietary fiber (with their derivatives), organic acids (which method was used) as well as sugar composition.

b.     Polyphenols content (12.95%) should be checked and recalculated for accuracy.

4.     Conclusion section is needed to improve with mentioned   issues. What is drawback in present research? Who will be interested on this research? What is its role in industry and research field?

               5.     The references should be revised carefully.

Author Response

Referee 1

We thank you for the revision and for your useful comments that contributed to ameliorate the overall quality of our paper. We answered below to your requests as well as we amended the manuscript as detailed.

Minor comments:

  1. The term in vivo, in vitro and scientific names should be written in Italic form through all the manuscript.

Ok, the terms have been written in italic, please see the revised text.

  1. Some sentences showed be revised/rewritten again to avoid repeated words or grammatical English language mistakes. For example, Page 3 line 5, 15, 25; Page 3 last paragraph line 3 (built correct to carried out); Page 4 subtitle 2.3.4  line 4 (integrated as was that…..correct to integrated as that)…………….and others

Ok, the sentences underlined by the referee and the remaining text have been corrected by an english mother tongue.

Major comments:

  1. The manuscript was not prepared carefully according to the journal guidelines. Where the orders of the main parts are not correct. It should be Introduction, Results and discussion, Materials and methods, Conclusion and references.

Ok the text has been ordered according to the guidelines of the Journal “Molecules”,

  1. Materials and methods part showed be enhanced by adding some details about:
  2. Nutritional composition, which components were determined? and give the complete citation for AOAC (Year) with methods numbers and also for ISO. The AOAC and ISO standard methods should be added in References list.

Ok the AOAC and ISO references have been added and methods for fatty acids, sugars  and organic acids determination reported. Please see the paragraph 4.3.5 Nutritional composition

  1. The authors should give the reason for Arabic gum addition in this stage (Page 3 line 7).

Ok. Arabic gum was used to stabilize the polyphenols during the spry-drying process. This explanation was added in the paragraph 4.2 on PBJ preparation.

  1. Page 5, subtitles 2.3.4 and 2.4.3. regarding the statistical analysis for anti-inflammatory and antioxidants; it should be removed from these subtitles then added in part 2.5.6. Statistical analysis.

Ok. The paragraph on statistical analysis relative to the in vitro and cell experiments has been moved to the Statistical analysis paragraph (now 4.5.6.).

  1. Results need some improvements
  2. Table (1), there are no details were mentioned in Materials and Methods part about the fatty acids, dietary fiber (with their derivatives), organic acids (which method was used) as well as sugar composition. 

Ok, methods have been added in the materials and method session.

  1. Polyphenols content (12.95%) should be checked and recalculated for accuracy.

Ok, the double digit is not suitable for this type of analysis which calculates the amount of the main polyphenolic classes using a common std for each polyphenol class. Based on the accuracy of the method and variability, the most correct way is to report a range of 10-13% instead of a double digit value. The value has been corrected in the text.

  1. Conclusion section is needed to improve with mentioned   issues. What is drawback in present research? Who will be interested on this research? What is its role in industry and research field?

Ok, the conclusion has been substantially improved by highlighting a drawback of the work and possible applications of the present research at scientific and industrial level.

  1.  The references should be revised carefully.

Ok the referees have been carefully revised.

Reviewer 2 Report

This work represents a potential use of bergamot juice as a functional ingredient, here represented as a by-product in the production of essential oils from bergamot peel.

However, the authors should provide more references in order to confirm this statement, i.e. that the peel is not considered as a by-product in the production of juice. 

Secondly, it is not clear enough why did the authors use both LC-MS and HPLC-PDA. Why couldn't the authors determine all components of interest solely by using LC-MS?

Some minor remarks also need to be addressed, for example:

- in vitro and in vivo, should be written in italic;
- in the abstract Nrf2 abbreviation was first time mentioned without the explanation of its meaning;

-authors state: small molecules and amino-acid betaines...aren't amino-acids considered small molecules?

Author Response

 Referee 2

We thank you for the revision and for your useful comments that contributed to ameliorate the overall quality of our paper. We answered below to your requests as well as we amended the manuscript as detailed.

This work represents a potential use of bergamot juice as a functional ingredient, here represented as a by-product in the production of essential oils from bergamot peel.

However, the authors should provide more references in order to confirm this statement, i.e. that the peel is not considered as a by-product in the production of juice. 

Ok an additional paragraph in the last part of the introduction together with references 15 and 16 have been added to confirm the fact that bergamot juice is a by-product of essential oil production, and poses a significant economic and environmental challenge. This aspect has been further underlined in the conclusion.

Secondly, it is not clear enough why did the authors use both LC-MS and HPLC-PDA. Why couldn't the authors determine all components of interest solely by using LC-MS?

An accurate quantitative analysis in LC-MS needs an isotopic dilution method therefore an isotopic std for each analyte is required. Moreover, the analytes, even those belonging to the same chemical class, can have a different ionization efficiency, so a std for each analyte is required. Hence, for a quantitative analysis in LC-MS, for each analyte, the std and the corresponding isotopic labelled derivative are required. This is not affordable both due to cost and the fact that most of the std and deuterated analogues are not commercially available. By contrast, the quantitative analysis by HPLC-UV-DAD is much simpler because it can be performed by using an external std method and analytes of the same chemical classes and characterized by the same chromophore and can be quantified using just one common external std belonging to the same chemical class. A paragraph reporting the approach for identification by LC-ESI-MS and quantitative analysis by LC-UV-DAD has been reported in the text (see paragraph 2.1)

Some minor remarks also need to be addressed, for example:

- in vitro and in vivo, should be written in italic;

Ok, please see the revised text

- in the abstract Nrf2 abbreviation was first time mentioned without the explanation of its meaning;

Ok, explanation of Nrf2 has been added (nuclear factor erythroid 2–related factor 2). See the revised text at page 7

-authors state: small molecules and amino-acid betaines...aren't amino-acids considered small molecules?

Ok, thank you for the comment. We have corrected the sentence in the abstract

Round 2

Reviewer 1 Report

The manuscript entitled “ Chemical, nutritional and biological evaluation of a sustainable 1 and scalable complex of phytochemicals from bergamot 2 by-products” was not completely revised by the authors and needs further improvments.

Some of the main comments:

Page 1, line 24: were should be corrected to was

Page 2, line 98, Page 3, line 99: The orders of the titles needs to be corrected.

Page 10, line 203: the order of the subtitle should be corrected.

Page 14, line 342: the order of the subtitle should be corrected.

Page 14, line 348: trasnportation.. (.) one of the should be removed.

Page 14, line 349: compound was dried? Which compound ? it should be the juice

Page 16, line 440: in vitro should be in italic form

Page 16, line 421-438: the AOAC and ISO was not mentioned in the references list and also the year of citation was not provided.

Author Response

Dear reviewer

Thank you so much for your careful revision and comments. Please note that most of the typos you have found (subtitle order, italic) were correct in the original text that I sent but then were changed during the formatting in the template of the  journal (probably it was made automatically). These typos do not depend on me, and I will carefully check the proof once I will receive it. All the others comments have been accepted.

The manuscript entitled “ Chemical, nutritional and biological evaluation of a sustainable 1 and scalable complex of phytochemicals from bergamot 2 by-products” was not completely revised by the authors and needs further improvements.

Some of the main comments:

Page 1, line 24: were should be corrected to was

In the sentence ”were” is referred to “values” which is plural. To be clearer we added the subject, “They”

Page 2, line 98, Page 3, line 99: The orders of the titles needs to be corrected.

In the original word document that I have sent the number were correct but I found that in the version formatted in the template of the journal the values were changed. I will check during the process of the paper.

Page 10, line 203: the order of the subtitle should be corrected.

Same problem as reported above, In the original text the numeration was correct

Page 14, line 342: the order of the subtitle should be corrected.

Same problem as reported above, In the original text the numeration is correct

Page 14, line 348: trasnportation.. (.) one of the should be removed.

Ok, thank you!

Page 14, line 349: compound was dried? Which compound ? it should be the juice

You are correct, I replaced “compound” with “juice”

Page 16, line 440: in vitro should be in italic form

 Again, this was correct in the original text

Page 16, line 421-438: the AOAC and ISO was not mentioned in the references list and also the year of citation was not provided.

Ok, we have added the title of the method and year and the web site of the organization from which the methods were downloaded was added in the text.

Reviewer 2 Report

The authors have improved the manuscript accordingly.

Author Response

Thank you so much for your revision